# Land Use Efficiency and Value Capture

Francesco Botticini [1],* and Armands Auzins [2]

1. Department of Civil, Engineering, Architecture, Land, and of Mathematics, University of Brescia, 25121 Brescia, Italy
2. Faculty of Engineering Economics and Management, Riga Technical University, Kalnciema Street 6-203, LV-1048 Riga, Latvia
* Correspondence: f.botticini002@unibs.it

**Abstract:** This article aims to relate intrinsic aspects of urban planning that are becoming increasingly important both within the international scientific debate and within urban planning tools. These aspects are land consumption and land value capture. Their centrality is given by the growing importance that in recent years have assumed the aspects related to the sustainable development of settlements. This article aims to summarize the main theories regarding the relationship between the efficiency of land use and the policies of capturing public value. The reference scenario is dictated by sustainable development policies that, if contextualized in the sphere of urban development, imply a rational use of resources to ensure the formation of resilient, safe, and inclusive settlements. This aspect introduces the existing link between urban form and sustainability. It is therefore understood how the achievement of the targets set at the international level are implemented in local urban policies. For this reason, many scholars have argued that the challenge of adapting to new pressures, such as climate change, necessarily means creating efficient urban settlements. The question becomes: which land use can be considered more efficient than the others? This article intends to answer this question by investigating the main theories that have tried to define the mechanisms and methods of comparison of different urban development scenarios. The article goes on to reconstruct the steps that have helped to evaluate urban development according to purely fiscal aspects up to the introduction of qualitative aspects related to public value. To do so, it analyzes the terms and tools related to the concepts of public value and the capture of public value and attempts to synthesize the main theories and research in the sector.

**Keywords:** land take; land use efficiency; terms and tools of public value capture

## 1. Introduction

The frame of sustainable land development embraces the issue of value capture. Historically, different assessment methods based on purely economic indicators have been applied to achieve the objective of evaluating the quality of urban growth and development. Urban growth has often been assessed on a purely quantitative basis. The criteria involved in this type of assessment typically has included the number of inhabitants, the number of city users, the expansion of urbanized land, and the economic value of the land. This latter parameter has often been used to decide whether to undertake an urban development operation.

However, theories have emerged in recent years that are slowly helping to change the sensitivity and orient the principles of urban development towards qualitative issues [1]. These theories are based on the principles that land rent has led to an imbalance in the use of resources, especially those related to the environmental sphere. One example is the urban taxation enforced in some European countries in the second half of the twentieth century, in which the urbanization of vacant areas was favored, as it allowed the collecting of charges that were used for the purposes of current expenditure and, therefore, contributed to increasing the municipal budget.

It is thus easy to understand the existence of link between how the soil is used and the type of urban taxation that is enforced in respective areas [1].

Issues related to sustainable urban development have forced other environmental, ecological, and social factors to be considered in territorial governance policies, such as the link between historical values and land values [2]. This has led to the gradual definition of public value—a parameter that can measure urban quality involving not only the economic sphere but also hedonistic aspects, e.g., the level of satisfaction of citizens living in a certain area.

The concept of public value is not entirely new. It has its origins in the value of land that in turn constitutes a next step compared to the purely economic concept of land rent.

The definition of the concept of public value leads to another important strand of scientific literature that relates to issues concerning the capture of public value (PVC), another aspect that is useful to consider within the mechanisms of territorial development.

In this article, the main theories underlying the concepts of land value (LV) and public value (PV) are analyzed. These concepts constitute a foundation of urban planning techniques. The concept of property, including real estate, has influenced philosophical debate since the XVIII century [3]. This concept began to have a weight in the scientific literature beginning in the 1990s, in particular regarding land value. However, within the last ten years, the debate has been animated considerably, and the number of publications concerning these terms has increased significantly.

This aspect allows to contextualize one of the basic problems of contemporary urbanism. With the aim of reducing land consumption and better managing of natural and environmental resources, is there a way to use land that maximizes its efficiency [4]? If so, how can the efficiency associated with each possible use be measured [5]?

To answer these questions, it is necessary to understand the connection between the use of the resource soil and the value associated with such use [6,7].

It is important to understand, therefore, the link between land use and the value that this use generates [8]. This value is not only economic, but it is also approached to the annuity, that is, to the gain because of an urban planning operation. The land value also has profound social and environmental consequences. To understand these concepts, it is important to think of the speculative phenomena that characterized much of the second half of the twentieth century that have led to the construction of dormitory districts with low architectural quality, which were often poorly equipped with minimal services. These areas have led to a very high consumption of resources [9]. Currently, within the planning process, attention is also paid to aspects such as the assessment of ecosystem services or social integration [9–12].

## 2. Data, Model, Applications, and Influences

### 2.1. Methodological Framework

This paper first identifies the definitions that are attributed to the concepts of value and property in the real estate sector in an international context, with main regard to the European scenario. The analysis of the definitions shows that the concept of value is closely linked to that of urban rent [13], a parameter that can measure the effectiveness of an intervention of urban transformation. Urban rent also has profound repercussions on the socio-economic matrix of a city [14–16].

The first step is the analysis of the main aspects related to the concept of value both in urban planning and in real estate economics. Second, these aspects are related to the most frequent theories of urban rent. Thus, a bibliometric analysis is made, which highlights the evolution that has had the most widespread terms, emerging from the studies in the first step. From this study it is viable to understand how and when the transition from a land-value-based analysis to a public-value-based analysis took place. This study made it possible to associate a series of keywords to key concepts and to understand which authors contributed most to the development of the research. This study mainly considers a European environment and perspective.

Bibliometrics is a science derived from infometrics [17,18] and is based on the implementation of bibliographical analysis. Bibliometrics often uses a software for statistical processing, e.g., to perform a systematic literature review. This science aims to quantify the qualitative characteristics of scientific publications to measure and compare how themes and concepts have evolved over time and determine the influences behind ideas and individual authors [17,19]. To obtain these results, scientific papers are analyzed by carrying out a three-level study. The first level is based on the definition of indicators to compare the impact of magazines, such as the impact factor and the normalized impact factor. The second level aims to analyze the indicators that characterize the impact of the authors; in this context the main parameter is the H-index. The third level compares individual articles in terms of their number of citations and co-citations [17,20].

The bibliometric analysis was carried out by filtering and analyzing the publications contained within the bibliographic database Web of Science. By setting a keyword, a time interval, and a geographical area, the articles were identified and analyzed in a statistical way to highlight key concepts, how many times these concepts are repeated, how the concepts have evolved, who the most relevant authors are, and how many publications the authors have written. These are just some of the parameters obtained by analyzing the metadata. The characteristic data that every article indexed in a scientific database must have include the title, abstract, keywords, year of publication, magazine in which the article was published, authors, and author affiliations.

### 2.2. Terms and Definition of Public Value

The concept of land value has historically asserted itself in the real estate sector. This concept was in fact linked to aspects such as income, that is, the gain that an operator could have from an urban operation based on the construction of residential housing—which has constituted the most profitable type of intervention since the second half of the twentieth century.

Subsequently, especially thanks to the affirmation within urban planning of tools to put into practice the characteristic themes of sustainable development (such as the Environmental Impact Assessment, the Strategic Environmental Assessment and, more recently, the reduction in land consumption, ecological balance, and urban regeneration), the concept of land value has been enriched with meanings and facets. In this way, environmental and social aspects have been combined with the economic value.

This is one of the fundamental steps that has contributed, first at the international level and then at the local level, to gradually introducing the theme of public value. It is understood not only as an economic gain resulting from an urban operation, but as a better quality of life in the urban environment following the implementation of a territorial development process.

The International Valuation Standards Company (IVSC), an entity that is part of the United Nations recognized bodies, has raised topical discussions surrounding land value. Since the 1990s, the IVSC has introduced several economic concepts to describe land value. The objective of this institute was to quantify the value that a resource, universally recognized as finite and limited (the soil or the land), assumes during a transformation process, and how this value can vary according to the different uses that are made of it (vacant land, building land, built-up land, agricultural land, etc.).

According to the meaning provided by the IVSC, land value is understood as a synonym for real estate property value, as the estimate that is made is strictly linked to market conditions and, more specifically, to the real estate market. The definition given by the IVSC takes inspiration from a concept introduced by Eckert in 1990. Thus, talking about the real estate property value, the concept of value in exchange is introduced, i.e.:

*"Market price Determined by the market."*

In this context, in 2001, the IVSC spoke of market value, circumscribing it in a context of demand and response (supply and demand). According to the IVSC, market value is defined as:

*"The estimated amount for which a property should exchange on the date of Valuation between a Willing buyer and a Willing seller in an arm's length transaction after proper marketing wherein the parties had each acted knowledgeably, prudently and without Compulsion".*

From these early definitions it emerges that, historically, the concept of value attributed to soil is purely an economic concept; soil was seen only as an object forming part of a transaction. This transaction could take place on two levels: horizontally between owners; or vertically between owners and public entities. This concept is the result of the model of the development of cities during the twentieth century, which expanded in a polypoid way on the territory following the main arteries connecting one settlement to another.

The soil has always been seen as a potential resource available to the private entrepreneur to increase profits through building operation, which were often speculative in nature. This led to the birth of increasingly peripheral suburbs, which became less and less equipped with services, and the classic dormitory districts. For this reason, the quantification of the value of the soil historically occurred according to monetary parameters.

Linked to this aspect is the concept of value in use which, since 1990, appears several times in the literature and is defined as:

*"The value refers to the current use of the property: often measured in the rent that the property can produce; valuations of value depend very much on the individual preferences of the owner or investors".*

The value in use is typically an expression of the first type of transaction—the horizontal one—which occurs between two individuals who treat at the same level. Linked to the vertical transaction is the concept of taxation (assessed) value discussed by Jurgenson in 2017 [21]:

*"The estimated value through the mass valuation; the cadastral value determined according to the common methodology, which in general prescribes the cadastral value base (base values and correction coefficients based on the analysis of market data)".*

These two definitions testify to the importance that is given to the ground in real estate transactions. For the private sector, it is more convenient to build on vacant land than to redevelop already urbanized areas. For this reason, the attention is paid to the value that the soil has according to its use. Estimation by the public sector is also part of this sphere, since the private sector benefits from real estate transactions and, the higher the value of the land after transformation, the greater its profit. The value of built-up land depends directly on the amount of volume that is built on a given portion of land. The estimation of this value is the basis of taxation mechanisms. The concept of taxation makes it possible to move from analysis related to the determination of value to analysis related to its capture by public administrations. The concept of land value capture, with its possible declinations, was defined in 2012 by Rachelle Alterman [3]. Alterman's analysis was a milestone in land value theory and land value capture mechanisms. Alterman defines the concept of land value as the value of which a private person benefits not directly due to his own efforts and initiative. Based on this consideration, the main theoretical foundations for justifying the capture of value from which a private individual benefits, by a public body, are presented. Finally, the main tools available to develop land value capture policies are illustrated. Alterman's thinking is also based on the analysis that van der Krabben and Needham made in 2008 [22], in which they discuss "instruments and public actions" and define the capture of value as:

*"A group of instruments allowing the direct or indirect recognition of the increase in the value of land and property as a result of public investment in transport infrastructure, so that they can be used to finance the activities responsible for the increase in value".*

These reflections are the basis of the concept of public finance theorized by Ingram and Hong in 2012 [17], and resumed by Smolka in 2013 [23], which is defined as:

"*Value capture focuses on realization as public revenue, e.g., through taxes, taxes or services in kind, a part of the increase in the value of the land resulting from these subsequent changes*";

and [24]:

"*Capture value refers to public recovery of land value increases. It is to mobilize for the benefit of the community in general some or all increases in the value of the land (unearned income or "plus valias") generated by actions other than direct investments of the landowner, such as public investment in infrastructure or administrative changes to land use rules and regulations. Although all these increases are essentially unearned income, value capture policies focus mainly on the increase generated by public investment and administrative actions, such as the granting of authorizations for the development of specific land uses and densities*".

From these definitions, some key concepts emerge that are linked to the theory of land value and land value capture; these terms refer to a series of benefits that a private investor obtains for merits that are not his but rather, are the result of planning decisions. For this reason, it is acceptable to capture all or part of this increase. Once the logical foundations of land value capture have been defined, Alterman moves to an analysis of the main terms (Figure 1) and tools (Figure 2) available to recover part of what is termed unearned increment [3,7,25]. Based on the various types of catch, Alterman identifies the terms that are used to describe the increase in value benefiting the private sectors in various countries where these techniques are implemented.

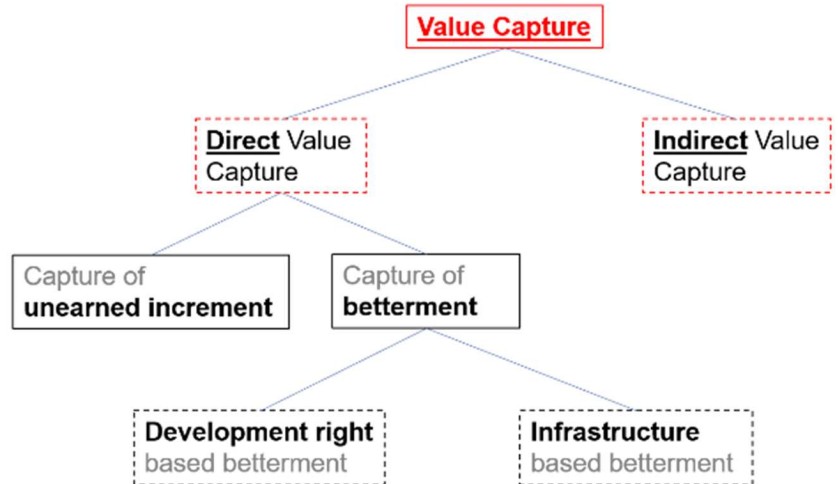

**Figure 1.** Main PVC terms in agreement with Alterman's classification.

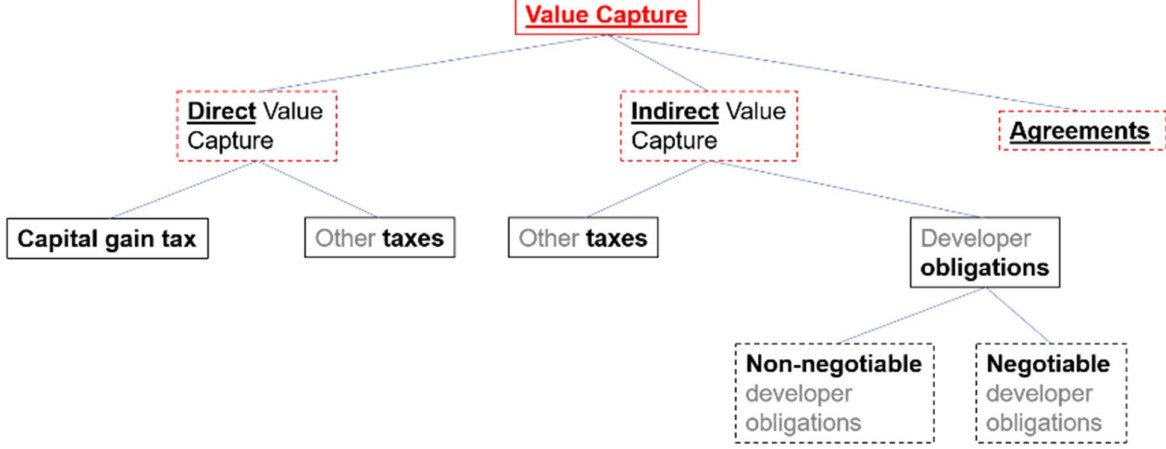

**Figure 2.** Main PVC tools in agreement with Alterman's classification.

*2.3. Tools of Public Value Capture*

Alterman divides the value capture tools into three categories: macro tools, direct tools, and indirect tools [3,25]. The first tools are instruments tied to the politics of the government of the territory. Accordingly, the territory is seen as a resource and commodity of exchange. Territory includes the nationalization of land and the replacement of private property. These two types of interventions are linked to communist traditions. After the collapse of communism these types have not found room for intervention. Within the macro-tools, one can also find the land banking that, although it has valid requirements from a theoretical point of view, also has significant gaps from the practical application standpoint. The fourth and last type of policy covered by this classification is land readjustment [3,25,26]. Although presenting valid ideas for its practical development, land development has never been systematically applied. One of the few European countries where these tools have been successfully applied is the Netherlands, where the value capture mechanisms have been used to finance the construction of public infrastructure [27,28]. The second group of tools consists of direct tools that aim to directly capture the value not created by private individuals, as this value belongs to the community [3,25]. These direct instruments aim to capture the so-called betterment in more detail. The value generated by public choices can have repercussions on private areas. Betterment can be generated either through the granting of building rights or through the creation of infrastructure [3].

When decisions on land use are made through the planning process, there are private individuals who benefit from these choices, as their properties are more centrally-located or better served by infrastructure. For this reason, some private land suffers an increase in value while others are disadvantaged by the choices of the plan. Alterman calls this phenomenon "windfall" when plan choices benefit a private individual, but "wipeout" when disadvantageous to a private individual [3]. This increase in value is not due to the efforts of the landowner and, for this reason, the surplus value that benefits his plot is captured; it does not belong to the individual operator, but to the community. The second strand of direct tools is the capture of the acquired increment, or "unearned increment" [3,7,25]. According to Alterman's analysis, only three countries have implemented direct value capture policies. These countries are the United Kingdom, Israel, and Poland [3,29,30]. The United Kingdom has been a laboratory of experimentation that throughout the course of the 1900s has seen the alternation of laws aimed at capturing the plus value, with poor results. The continuous alternation of Labor-led governments and Tories-led governments has resulted in the various laws put in place by one faction being systematically erased by the opposite faction. For this reason, the laws have never been able to produce tangible effects [3].

In Poland, however, the tools for direct value capture have had a wide theoretical diffusion but little practical application [25]. The third state in which there is evidence of direct capture tools is Israel. As a British protectorate, Israel has seen a profound influence of British laws on their tax system, and urban taxation has also suffered. For this reason, there is a long tradition in the application of land value capture policies that have resulted in two different tools: a tax of 50% of the increase in the value of the real estate and a tax of 25% on the increase not acquired [3,31,32].

Direct value capture tools also include negotiation agreements, as they aim to directly capture the additional quality that is induced in the territory by a transformation [26]. The agreements take place between the public body and private operators and are part of the group of direct instruments because the developer of the plot can directly benefit from planning choices that, in relation to the strategies, recognize a certain portion of territory as strategic for development. For this reason, the private operator is explicitly asked to contribute to the transformation of public land too. This is accomplished by returning to the community part of the surplus value from which it has benefited. The last group of value capture tools are indirect vale capture tools. These tools are often confused with direct instruments in that they aim to capture the same thing but, unlike direct instruments,

have different logical foundations [7]. According to Alterman, there are two categories of indirect value capture tools: developer obligations or other types of taxes [3].

*2.4. PVC as Tool to Assess Urban Efficiency*

Alterman's theories are the basis of the research of many other scholars who have discussed the mechanisms of generation and capture of value. In the Anglo-Saxon world, the theme of the surplus value generated by the infrastructure of the territory has always been strongly felt. The city of London has for some time conducted a study regarding the increase in value that private areas undergo in relation to the proximity of subway stops [33]. This aspect falls within the concept of infrastructure-based betterment. The final report of this study emphasizes the importance of the proximity of an infrastructure that makes the territory accessible for the increase in the value of a plot of land. The closer someone is to the access point to the infrastructure, e.g., a subway stop, the greater the benefit (windfall or betterment) is and the more fiscal tools that are available to the public body to capture that increase. These theories are also confirmed by the studies of Medda, who analyzed the relationship between the value of private areas and the proximity of infrastructure related to public transport [34]. This concept has been deepened by van der Krabben and other authors of the Dutch school and is a very popular theme in the United States, Australia, and China. According to scholars, proximity and accessibility to public transport contribute to making areas more attractive. As one example, as the price of individual apartments increase as the entire value of an area increases, private individuals are willing to spend more to buy them [6,34–36].

As Havel concludes, this relationship is bidirectional [25]. On one hand, the private individual benefits from public initiative, but on the other hand, it is also the public that, capturing surplus value, manages to finance the infrastructure or process that is responsible for generating this value [22,26]. For this reason, the capture of value can become a tool in the hands of public bodies with which they can find funds for the construction, maintenance, and enhancement of infrastructure that relate to public transport and soft mobility [6,26,33,37].

Hendricks focuses his analysis on the theme of capturing what is called unearned increment [7]. His analysis focuses more on the right-based development betterment that is the surplus value of which private plots benefit because of the extension of building rights by the public body. According to Hendricks, it is necessary to analyze the mechanism by which a surplus value is formed, understand who the involved actors are, and identify the resources used to obtain the surplus with the aim of understanding who has the right to capture what [7]. With this aim, Hendricks identifies three phases within the territorial development process: predevelopment, development, and post-development. These three phases are divided into five steps. The predevelopment phase, according to Hendricks, is formed by an initial state characterized by a certain types of land use, while the second step is the extension of building rights in which there is a transition from a state of fact to a rule of law. The development phase consists of two steps: the construction of internal and external infrastructure; and the implementation of the development project of the sector in in which the transition from the initial land use to the final one occurs. The last phase is that of land use after the conclusions of previously planned development [7].

The objective of this analysis is to understand what resources are used and by whom they are used in different phases of territorial development to obtain certain benefits. The aim is to understand if there are any unaccounted-for benefits and who has the right to catch them. This type of analysis opens developments related to the efficiency of land use [5,14,38], and also allows the comparison of resources used and the benefits obtained in different scenarios. It can then be determined what is "the highest and best land use" [4].

These studies are included within the framework of direct value capture mechanisms. Other research on indirect mechanisms exists. Van der Krabben and Oppio focused on developer obligations as tools to negotiate with private individuals and then capture, through their intervention, part of the surplus value [26,39]. Oppio argued that indirect

instruments can be divided into two categories: urban development agreements and fiscal instruments [39]. According to Oppio, the former has been widely applied in Lombardy (northern Italy). A fundamental characteristic of indirect instruments is that their application, in addition to the laws in force, depends strongly on individual cases and therefore, have a very high variability [39]. Van der Krabben is of the same opinion and argued that developer obligations strongly depend on specifications to be determined on a case-by-case basis [26]. According to van der Kabben, there are two types of developer obligations: negotiable developer obligations (NDO) and non-negotiable developer obligations (N-NDO) [26]. Moreover, van der Krabben distinguished between land value and public value by circumscribing the first term to the economic value of the soil during a transformation process. The second term—public value—in agreement with Alterman's theories, is a more generic term. In addition to economic value, it also includes other aspects [3,26]. According to Alterman, the term public value can be used to describe the various declinations that land value assumes in different countries and jurisdictions [3]. The concept of public value is also taken up by Hendricks [7], who combines the concept of public value capture with that of public finance. This explains that the capture of plus value can be a tool in the hands of the administration to make sure that the private sector contributes to public finance [7].

### 2.5. PVC, Urban Efficiency, and Ecosystem Services

After analyzing the theme of soil as an environmental resource from which benefits can be derived, it can be concluded that it is necessary to find a common system to quantify the externalities that nature offers to the urban environment. For this reason, many studies are focused on the quantification and monetization of ecosystem services.

Although these studies have a broader scope than the purely economic ones typical of the analysis of rent, currency is still the main benchmark for the assessment of planning choices. The externalities are quantified in terms of expenditure to answer questions such as the cost of the implementation of the technologies for the reduction in atmospheric pollution, and of the obtained benefits (always in monetary terms).

The issue of public value has focused more on aspects concerning the governance and management of the territorial development process. Van der Krabben's example is emblematic, linking the concept of public value to that of value capture.

There is the matter of public value capture as a set of urban policies justifying the fiscal mechanisms that allow public bodies to recover part of the resources invested to finance the process. A theme that is closely related to value capture is that of value creation itself.

These are the premises for research lines such as the value-led planning approach (VLP). Public value studies focus on how resources can be used as efficiently as possible to achieve the greatest benefits at the lowest cost.

While economic assessments are essential, they are also accompanied by studies on more qualitative aspects.

The well-established theory that public value capture is a tool to ensure that private resources are used by the administration to contribute to the quality of public areas is the basis of the concepts of urban regeneration and sustainable development. Auzins discusses possible development in the light of a value-led planning approach. The analysis of the mechanisms of public value capture leads the public body to evaluate, within the planning process, different possible scenarios through forms of public partnership. The different actors involved in the process contribute to the efficient management of resources to pursue the objectives of environmental and socioeconomic quality [4]. This research strand includes studies by Gomez-Baggethun on the evaluation and monetization of ecosystem services [40–42]. The quantification of ecosystem services is another method to evaluate the quality of a land use transformation from an ecological and environmental point of view. Therefore, this aspect can be taken as an indicator to measure the quality of an area. In Italy this theory has been taken up by the Major Institute for Environmental Protection and Research (ISPRA), which has developed a methodology for the monetization of ecosystem

services with the aim of making comparable changes in land use with different effects on the built environment [43].

This last aspect introduces the issue related to the link between urban development and ecosystem services.

The relationship between ecosystem services (ES), land use change, and public values has been widely discussed in the recent scientific literature [44,45]. Economies depend on ES, but only recently have ES values been incorporated into economic and political decision making [44]. Exploring ES deficits of European cities, Elliot et al. (2022) provided arguments that urban sustainability strategies should complement local restoration with changes to local consumption alongside promoting remote ecological restoration to tackle the multilevel environmental impacts of cities [46]. Thus, the urban ES strategies should focus on altering consumption patterns. Stępniewska et al. (2022) referred to the transdisciplinary approach that lies at the core of the ES field and provides a unique opportunity to develop scientific foundations of the concept and foster the integration of nature's values into governance processes [47]. Thus, by using an interdisciplinary approach, the differences in the methods used, the concept of valuation, and the specificity of policymaking have been highlighted by several scientific contributions.

Various aspects of ES in relation to using public and landscape values to support a decision-making process have been discussed in the scientific literature in recent years. Maund et al. (2020) focused on a common international classification of ES and frameworks to capture social values [48]. These authors emphasized the need to consider how society views the services derived from nature and reflect this in frameworks to ensure ES approaches are effective, transparent, and widely supported. Promoting ES-based actions, Khoshkar et al. (2020) provided a national (Swedish) overview of the integration of ES in a local planning context [49]. Lourdes et al. (2022) proposed multiple urban ES models and a multicriteria analysis, thus supporting planning for green infrastructure [50]. Custodio et al. (2022) capitalized on stakeholders' preferences for prioritizing ES for marine management [51]. Assessing the impact of urbanization and the capacity of ecosystems in the Lisbon Metropolitan Area, Mascharenhas et al. (2019) suggested that the effects of urban development on land take are positive for a "compact city" and negative for an "urban sprawl" pattern, even for opposite demographic developments [52]. Berglihn and Gomez-Baggethun (2021) revealed the fundamental importance of Oslo's urban forests and its ecosystem services for the well-being of local inhabitants [53]. Chen et al. (2022) assessed regulating urban ES and suggested a more comprehensive real estate pricing. [54]. This contributed to further environmental urban planning and provided a basis for the implementation of relevant policies, thus moving beyond the traditional pricing methods. Providing a meta-regression analysis of the economic value of grassland ES in China, Liu et al. (2022) contributed to a better accounting of the services, which can significantly facilitate land use decision making for sustainable ecosystem management [55]. Performing a cross-site analysis of perceived ES benefits in multifunctional landscapes, Fagerholm et al. (2019) addressed conceptual confusion in the ES framework and presented arguments regarding the links from services to benefits, and from benefits to different types of values [56]. Analyzing urban expansion, land-use transformation, spatial-environmental impacts, and trends and implications in major metropolitan regions of Ghana, Asabere et al. (2020) argued for the need for integrative urban growth management strategies that bring together spatial planning and environmental resource governance to avert the negative consequences on the natural environment of an unfettered urban expansion [57]. Demonstrating a spatial prioritization approach of biodiversity and involving stakeholders, Vaz et al. (2021) contributed to the integration of conservation targets and ES in landscape spatial planning in Portugal [58].

Empirical studies worldwide have demonstrated the effects of land-use change on ES. The growing interest in ES is mainly related to land-use change. Temporal changes in overall ES capacity indicated an increase in this metric in Lithuania over the last two decades [59]. Given the observed dynamic context of land cover, the structure of ecosystem

services may face potential threats from land-use change due to urban development and agricultural activities.

Li et al. (2022) provided arguments that land use conversion significantly affected ES in the China–Mongolia–Russia economic corridor from 1992 to 2019. These authors provided new knowledge that enriches the understanding of ES in transnational areas and promotes balancing the relationship between ecological conservation and socioeconomic development in particular areas [60]. Applying an ES-based strategic environmental approach, Nijhum et al. (2021) evaluated alternative land-use scenarios in urban ecological areas in Canada [61]. These authors discussed the improved integration of ES valuation in strategic land-use planning and the translation of complex ES concepts into practical directions for urban residents and for those responsible for land use policy and planning decisions. As research on the relationship between ES and urbanization has been drawing attention in recent years, Wang et al. (2022) specified the impact of urbanization on ES and highlighted the importance of spatial scales in a mega metropolitan area of the Pearl River Delta region by studying the relationship between urbanization and ES [62]. Exploring, how ES drives urban growth, Pan et al. (2021) tested the socio-ecological model in Stockholm County and suggested a place-specific nature-based strategy for addressing the heterogeneous spatial relationship between ES and urban development [63]. Providing a framework and empirical evidence from China regarding the effect of an urban–rural construction land transition on ES, Xing et al. (2022) proposed specific policy recommendations for sustainable urban development in the regions of China [64]. The proposed framework also provided effective guidance for urban planning in other rapidly urbanizing cities across the world.

## 3. Discussion and Prospects

When speaking about the value of the soil (land), it is also necessary to consider that the soil has more than just its economic value. Land also has a humanistic value, which is its characteristics that have allowed the development of settlements and community life. This aspect is clear to see in cases where urban development was based on the enhancement of historical parts of the settlement such as urban cores [2]. As Alterman mentioned:

> *"The earth is essential for human physical survival, provides the essential context for social and political life, often incorporates symbolic or religious values, is largely irreplaceable, every place is unique, and the supply of land is almost over"* [3].

This concept was also adopted by Hendricks who claims that:

> *"Land and its value play a crucial role in activities and social development"*. [7].

From these considerations emerges clearly an aspect related to the symbolic value that human beings attribute to the soil: the fact that, as Alterman adverted, is a finite and finite resource as the soil with the characteristics suitable to accommodate the development of settlements. Just think of the data reported by the Habitat III conference that took place in Quito (Ecuador) in 2016. Data showed that, currently, cities occupy only 3% of the land. These considerations are also the basis of the theories of Busi and Tira who argue that, like other natural resources, the soil must also be protected with a view to sustainable development [11,65]. This concept has led urban planning issues to evolve and for this reason it is better to talk about the government of the territory as the discipline includes, in addition to aspects related to the quantitative development of cities [9]. This concept has marked urbanism throughout the twentieth century. The government of the territory also includes qualitative aspects such as the provision of services and infrastructure, also assess the impact of plan choices on natural resources such as air, water and, of course, soil [1]. For this reason, it is necessary that the evaluation of possible development scenarios becomes part of the planning process. This analysis allows, in fact, to evaluate possible land use solutions and associate to each scenario the resources that are needed to obtain certain benefits" [1,4,5,14,38].

The above aspects have contributed most to the transition from land value to public value because the urban changes are no longer measurable in economic and quantitative terms only; it is also necessary to consider qualitative aspects that contribute to creating better living conditions in cities [1]. These aspects can make settlements more attractive and responsive to new stresses such as increasing migratory flows or climate change [66,67] in accordance with the main international agendas, namely, the 2030 Agenda for Sustainable Development drawn up by the UN and the derived Amsterdam Pact drawn up by the European Union.

It is no coincidence that in recent years, much research in the European field has focused on defining theories for the definition of a methodology that can quantify aspects related to quality of life in the urban environment. Some examples are the research of Gomez-Baggethun [41,42,68] on the monetization of ecosystem services, or those of Auzins [21] and Pelorosso [69–71] that have developed in parallel and distinct ways, an analysis of the relationship between value created and planning tools.

Even if only the purely economic aspects linked to spatial development are considered it is important to include the natural and social resources involved in the process. For this reason, it is necessary to split between two mechanisms of urban operations because the analysis of the effects on the urban system will differ depending on which mechanism is used.

The first mechanism, territorial development, involves urban planning operations in an extra-urban environment. The second, on the other hand, considers the processes of redevelopment of the built environment. The operations in the suburban area are those typical of the economic boom that, between the 1960s and 1980s, had a strong pull from the real estate market. The second mechanism is more recent and is the result of several EU directives and international targets that have introduced the concept of zero land consumption. Within the urban and regulatory debate, issues related to urban regeneration and the enhancement of heritage have become increasingly important, also through energy efficiency systems and the adaptation to new performance standards.

Another subdivision that needs to be made when analyzing a land development process is that concerning role of the different actors. Depending on the type of actions and mechanisms that are created between private operators and public bodies, one can speak of passive mechanisms or pro-active mechanisms [72]. One of the major authors who introduced these concepts was Van der Krabben, who described the difference between land value and public value by putting forth that the first is based on a quantitative analysis of resources while the second introduces new aspects [15,73].

The considerations made so far can be implemented in the analysis of the steps of the value creation defined (and previously described in this paper) by Hendricks. At this juncture, one can see how Hendricks approaches spatial development by describing the process of urbanization of vacant areas on which building rights are lowered. This mechanism is undoubtedly the most widespread and the one that has been most applied within the process of settlement growth. However, the problems related to land consumption have introduced limitations within the land-regulation system, for example, the objective of net land consumption of zero, otherwise known as No Net Land Take, [74] by 2050.

Thus, the challenges related to this new objective are leading to a change of mentality with profound repercussions for territorial governance. A striking example is seen by looking at the most recent regulatory plans in which many areas of transformation in the suburban environment are cut and increasingly assume a weight in the areas of urban regeneration [75].

Within this framework, the introduction of a VLP approach based on consolidated new knowledge from stakeholders' experiences and empirical evidence will help better understand and guide the relevant processes and their effects in specific territories based on (1) the identified values as an outcome of experts' work; and (2) the attitudes from stakeholders' preferences concerning these values [76].

Another author who has analyzed different mechanisms of value capture in various states is Munoz-Gielen [77–79]. His studies showed that there is no common European guideline but rather each country, in relation to its specific history, has consolidated different techniques in which the relationship between the actors varies significantly therefore alter the mechanisms underlying the territorial development process. In the study of Munoz-Gielen, the urban planning process is broken down with the aim of highlighting, in the various steps, the benefits obtained (understood as territorial endowments, infrastructures and services) and to the resources used to obtain them belong. Moreover, processes in different countries with different planning systems were compared. The analysis showed that, depending on the planning systems in force, the entity that must implement the territorial allocations varies [79]. This further testifies to the fact that the capture of value largely depends on the specifics of the process and are therefore defined on a case-by-case basis [3,80]. It emerges, therefore, that the existing planning system is not fundamental, but rather, comes from the ability of the public body to negotiate with private investors and develop partnerships that can promote the economic sustainability of the process.

Less scientific contributions have explored the relationship between ES and value-capture policies and specific instruments in recent years. Exploring governance rigidity, industry evolution, and the value capture in "platform ecosystems", Uzunca et al. (2022) showed how enterprises can capture above-average rents by controlling hard-to-replace segments [81]. Their study and proposed framework aimed to understand the combined effects of initial conditions and governance rigidity as key drivers of a platform owner's ability to capture rents. Morgan et al. (2022) examined several mechanisms that exist to create incomes from the forest through more sustainable activities that recognize and seek to capture forest ES benefits beyond timber [82]. They provided arguments that, while the mechanisms recognize multiple ES, they struggle to demonstrate their value, and thus, ineffectively capture them in forest management and income generation for forest stewards. The authors also provided guidance for a more comprehensive valuation of forest ES, inclusive of ecosystem integrity, that enables the sharing of benefits. Their approach considers an integration of all the benefits and the beneficiaries. Therefore, the agreement on the values and equitable sharing of the benefits through participatory planning and governance is essential. Röhrig et al. (2020) recently assessed if farmers can translate the ecological value into economic benefits [83]. The experience of value capture of ES from agricultural systems showed that designing public payments to reward the provision of ES directly could make this kind of farming more attractive and encourage wider application. Grimsrud et al. (2020) identified three prevailing stakeholder perspectives in the climate—forest debate using the Q method that demonstrates a way forward in ecological economics to better capture representative values and perspectives in ES management and help design climate and environmental policies with greater acceptance [84]. As traditional agricultural landscapes provide cultural ES to human communities, Borrello et al. (2022) provided findings on agriculture landscape certification as a market-driven tool to reward the provisioning of cultural ES. Their study proved that these landscapes are not necessarily bound to market failure. However, the costs faced by farmers to maintain them can be rewarded through the market [45]. A recent study by Wu et al. (2022) enhanced the understanding about this relationship, which is relevant for the conservation of plant biodiversity, a functional enhancement of ES, and the sustainable development of urban forest ecosystems [85]. Zhao et al. (2022) discussed distinctive hydrological responses following from different protection policies that were influenced by both climate change and land-use change [86]. Topical studies from India and South Africa have focused on different aspects of regulation and valuation of ES in particular scales [87,88].

## 4. Conclusions

With reference to the considerations made in this paper, the processes of spatial development can be evaluated based on two aspects: the rent that can be obtained from the urban operation; or the plus value that is generated because of the transformation.

Considering only rent as a parameter can give rise to speculative phenomena—which characterized much of urban development in the second half of the last century. The analysis of the plus value is more complex as is composed qualitative parameters that are not directly measurable in addition to the economic aspects.

This article mainly focused on the link between land use and value creation in the European context. However, there are experiences related to this issue that cover other countries such as China. The most relevant studies of those countries were cited in this paper as transversal experiences of implementing ecosystem services in land management. One possible development of this research may be the extension of the methodology developed in this article to case studies outside of Europe. For one, the Chinese experience can be deepened as the literature regarding how these topics are dealt with in that country is very extensive. Countries in African and Indian regions can also be included in the analysis, as they largely represent cases of developing countries.

**Author Contributions:** Writing original draft: F.B., supervision: A.A.; terms and tools of PVC chapters: F.B. and A.A.; bibliometric analysis: F.B.; ecosystem service chapter: A.A.; conclusion: F.B. and A.A. All authors have read and agreed to the published version of the manuscript.

**Funding:** This research received no external funding.

**Institutional Review Board Statement:** Not applicable.

**Informed Consent Statement:** Not applicable.

**Data Availability Statement:** Not applicable.

**Conflicts of Interest:** The authors declare no conflict of interest.

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
