# Peer review of "Land Use Efficiency and Value Capture"

_encyclopedia, doi:10.3390/encyclopedia2040134_

Round 1
Reviewer 1 Report
The manuscript is interesting, however I consider that it should address some comments before continuing with its processing, which is indicated in the attached file

Author Response
thanks for reviewing my manuscript. the reply to your comments is in the uploaded file.

Reviewer 2 Report
I recommend it for publication with minor revision. I suggest the authors to organize the structure of this manuscript better since some sections are too long. For instance, the introduction should be divided into several sub-sections with numbers. Similarly, section 2 "Data, Model, Applications and Influences "should also be divided into several sub-sections.
Author Response

(The authors gave the same response as above.)

Reviewer 3 Report
The authors cover a very interesting topic to understand the relationship between the efficiency of land use and the policies of capturing public value. The manuscript is well written with important and concise discussions. When I started reading this manuscript I thought authors might have covered the subject in global content as they stressed the content as a global issue, but in the manuscript, authors cover mostly over European regions. Since land use change is happening over the years and it will be continuing specifically over large developing countries like China, India etc. But authors did not touch anything over these regions. There are plenty of works on the land use change and their impact on various aspects are discussed in recent years. So, I suggest the following points to be incorporated before it gets published.
1. Include more literature to highlight this content across the globe especially the large developing countries like China, India etc.
2. The manuscript is not structured well. Please provide more heading and/or subheading to understand the discussed aspects well.
3. I strongly suggest a discussion on the “land use changes and their impacts across the globe”
4. Number of literatures looks less for a review paper.
5. Separate out the conclusion and provide it more briefly.
6. Reference and citation style: Follow the MPDI guideline
With reasons above, I am afraid that I cannot suggest the proper recommendation unless the authors address the above critical issues. Therefore, my recommendation is "The paper needs substantial revisions prior to rendering a judgment".
Author Response

(The authors gave the same response as above.)

Reviewer 4 Report
- incorrect in-text citations: missing a year of reference publishing,
- figures no. 4 - 6 could be smaller from point of view the presented content,
- deeper analysis of ecosystem services concept would be appropriate
Author Response

(The authors gave the same response as above.)

Reviewer 5 Report
General overlook:
This paper had an interesting and timely concept. The research was extensive and large-scale. Environmental Decision Makers will find the data obtained from the research to be particularly useful. Yet, the document needs major revisions. Details are provided. For discussion, authors should refer to international literature to provide some examples, such as estimating property rights and land policies. The literature part of the article is very outdated, even though the article is well organized. The article needs to be better grounded in recent works, especially in literature from other regions.
Specific remarks:
1. English needs to be improved. I strongly recommend that authors use a professional language.
2. In the manuscript, it is recommended that professional terms be unified. A lack of clarity in the purpose of the manuscript was evident in the introduction. The formatting of references in the text should be consistent, please check.
3. This manuscript contains some spelling errors, so it should be double-checked. In the introduction, it is difficult to understand the value of the solution and the innovative nature of the methodology used and the value of the solution.
4. A number of the statements in "Data, Model, Applications and Influence" appear reasonable after reviewing the results. Those conclusions align with the research objective. Based on long-term field experiments, a relevant study would investigate the long-term effects on ecosystem properties. Would this be the intention of the authors? Are there markets that we can trust? It is important to know the study's purpose.
5. In this paper, evidence is presented to support a particular viewpoint within the field viewpoint in this field. Thus, you should place a great deal of emphasis on the data that supports or refutes your position in your paper.
6. The experimental method used to obtain the data should also be explained. Review papers focus on the interpretation of primary literature on a particular topic. Make your own conclusions about the meaning of the papers you have read by reading several original research articles on the same topic.
Constructive feedback:
It might be an interesting paper for scientists deeply interested in setting the boundaries of market-based instruments in ecosystem services government.
The abstract provides a summary that is adequate for the manuscript. There is a concise introduction. There is a clear statement of why the study was performed. Based on a review of published literature, the authors justify the study. The authors describe the methodology appropriately. The methods can be reproduced by another investigator. The results are clearly explained and reasonable. The conclusions are supported by the data. A number of classifiers are applied to data from multiple sources in the paper. In another part of the method, information is also gathered ad hoc. In addition to the feature values of samples from various classes, it appears to have many attributes. Despite the ability to capture a lot of information, dimensionality may be an issue due to the identification of multiple conditions. Segmentation results with varying parameters produce huge amounts of variables. I believe the paper presents a feasible solution that can be very useful. You failed to consider the implementation of value capture policies and tools for urban development policy focus, which is a pity. Innovative techniques and practices for land management are needed for urban dynamics. Future research on spatiotemporal patterns and sustainable development will add criteria for assessing land ecosystem health to existing criteria and result in spatial object-based image analysis analyzing the distribution pattern of recorded diversities in the region to enable precise mapping and monitoring. Is this article similar to other publications on the same subject or does it differ?
Summary:
A valuable resource for academics. This article engaged me in addition to providing several types of data. The article needs to be revised in a few places. A discussion of opportunities and challenges should be included in the introduction regarding the estimation of and-use regulations and property using earth observation data as well as driving climatic forces, in general. Literature references should also be included.
Author Response

(The authors gave the same response as above.)

Round 2
Reviewer 1 Report
the manuscript was improved and is now ready for publication
Author Response
thanks for your help in the development of the manuscript and for the positive evaluation.
Reviewer 2 Report
The authors' effort for substantially enhancing quality of this manuscript is highly appreciated. I believe this manuscript meets the criteria for publication in its present form.
Author Response

(The authors gave the same response as above.)

Reviewer 3 Report
I appreciate the authors' effort. However, it seems the authors have not gone through my comments carefully. I didn't see any substantial improvements in the revised manuscript. Authors replied that they improved the introduction, where? I could not find it. Authors said they discussed and compared their results with previous studies, where? I did not recognize it. So I must suggest authors to go through all the reviewers' comments and reply POINT-TO-POINT with line numbers where did they change.
Author Response
thanks for your help in the development of the manuscript
a point to point reply to your comments is added

Reviewer 4 Report
Thanks to authors for cooperation and processing of interesting article.
Author Response

(The authors gave the same response as above.)

Reviewer 5 Report
A few things have been fixed and some problems have been resolved. Is my opinion the same as yours?
Could you please answer something for me? You should also try synthesizing sentences and general methods since THIS WORK IS IMPORTANT. Check that you have not made any mistakes when getting published. Would you mind checking for typos?
Author Response
thanks for your help in the development of the manuscript and for the positive evaluation.
an English check was done. now some typos are corrected and the structure of long sentences is reduced.
Round 3
Reviewer 3 Report
I appreciate the authors' Ponit-to-Point reply. However, authors have not gone through my comments carefully. I didn't see any substantial improvements in the re-revised manuscript. Authors replied that they improved the introduction, but they did not improve anything. I don't agree with authors' reply that they have highlighted the background information in line 240-250. It is a review paper. So introduction must be enriched with suitable background information, which I could not find. Authors said they discussed and compared their results with previous studies, but where? I did not recognize it. So I must suggest authors for the final time to go through all the reviewers' comments and reply POINT-TO-POINT with line numbers where did they change.
Author Response
dear reviewer, thanks for the time you spend reviewing our paper. we are sorry that you don't agree with our implementation. anyway, we go through your comments and add some new references about recent research dealing with the issue of land development and value capture.
as is explained in the paper the focus of the research is the link between these two aspects mainly in the field of European experiences. we added some experiences from other foreign countries too. this is to give a brief focus on extra-European experiences but this is not the main point of the research.
we think that the number of references is enough because there are almost 90paper cited. some of them are very recent while others are quite old but are still important and are the basis of the research and let to understand the definitions and the contexts at the base of the article.
we hope you will agree with the last development of the paper.
Reviewer 5 Report
This paper has an interesting and timely concept.
Author Response
dear reviewer, thanks for the time you spend reviewing our paper.
we did an English check and we hope that now the paper is suitable for you